# Epigenetic Regulation of Human T-Cell Leukemia Virus Gene Expression

**DOI:** 10.3390/microorganisms10010084

**Published:** 2021-12-31

**Authors:** Lee Ratner

**Affiliations:** Department of Medicine, Division of Molecular Oncology, Washington University School of Medicine, Box 8069, 660 S Euclid Ave, St. Louis, MO 63110, USA; lratner@wustl.edu; Tel.: +1-314-362-8836; Fax: +1-314-747-2120

**Keywords:** HTLV, ATLL, CpG methylation, histone methylation, CTCF, boundary element

## Abstract

Viral and cellular gene expression are regulated by epigenetic alterations, including DNA methylation, histone modifications, nucleosome positioning, and chromatin looping. Human T-cell leukemia virus type 1 (HTLV-1) is a pathogenic retrovirus associated with inflammatory disorders and T-cell lymphoproliferative malignancy. The transforming activity of HTLV-1 is driven by the viral oncoprotein Tax, which acts as a transcriptional activator of the cAMP response element-binding protein (CREB) and nuclear factor kappa B (NFκB) pathways. The epigenetic effects of Tax and the induction of lymphoproliferative malignancy include alterations in DNA methylation and histone modifications. In addition, alterations in nucleosome positioning and DNA looping also occur in HTLV-1-induced malignant cells. A mechanistic definition of these effects will pave the way to new therapies for HTLV-1-associated disorders.

## 1. Epigenetic Dysregulation in Virus-Associated Cancers

The Greek prefix “epi” means “over, outside of, around” [1]. Thus, epigenetics refers to features that are “in addition to the traditional basis of inheritance” [2]. In other words, epigenetics refers to changes that affect gene activity and expression that do not involve a change in the nucleotide sequence [3]. Examples of mechanisms that produce such changes are DNA methylation and histone modification. Waddington coined the term “epigenetic” in 1942 before the physical nature of genes was known [4]. Holliday defined epigenetics in 1990 as the “study of the mechanism of temporal and spatial control of gene activity” [5].

All major classes of cancer-causing agents are associated with epigenetic re-programming. This includes cancers caused by chemical carcinogens, hormones, heavy metals, radiation, viruses, and bacteria [6]. There has been a great deal of research on alterations in the epigenetic landscape induced by tumor viruses and their transforming proteins [6]. These include studies of human tumor viruses, such as Kaposi sarcoma herpes virus (KSHV), Epstein–Barr virus (EBV), human papilloma viruses (HPV), hepatitis viruses B (HBV) and C (HCV), Merkel cell polyoma virus (MCPyV), and human T-cell leukemia virus (HTLV).

## 2. HTLV and ATLL

Human T-cell leukemia virus type 1 (HTLV-1) is the first discovered human retrovirus, and it has infected 15–20 million individuals worldwide [7]. HTLV-1 is the cause of adult T-cell leukemia/lymphoma (ATLL), a CD4 + CD25 + CD62L + FoxP3 + CCR4 + CADM1 + effector memory T-cell lymphoproliferative disorder that accounts for 5% of T-cell lymphomas and leukemias [8]. It utilizes signaling pathways activated in a wide range of lymphoproliferative diseases and provides a relevant model system for deciphering the principles of cancer biology [9]. Infection also causes inflammatory disorders, e.g., HTLV-1-associated myelopathy (HAM), pneumonitis, uveitis, arthritis, and myositis [10]. Most individuals who develop ATLL in adulthood were infected as infants via breast-feeding [11]. It remains unclear as to why ATLL requires decades to manifest; why only 5–10% of infected individuals develop disease symptoms; what the nature of virus expression and replication is during the lengthy period of clinical latency; and how HTLV-1-associated diseases can be prevented and treated.

## 3. Tax and HBZ 

HTLV-1 is a plus single-strand RNA-containing δ-retrovirus that expresses the structural proteins Gag and Env, as well as enzymatic proteins, reverse transcriptase, protease, and integrase (Figure 1) [12]. Like other δ-retroviruses, HTLV-2, and bovine leukemia virus (BLV), HTLV-1 expresses Tax, the potent transcriptional activator, and Rex, a regulator of the export of incompletely spliced nuclear RNAs as well as other “accessory” proteins that regulate aspects of virus replication and immune evasion (e.g., p12/p8, p30) [13]. In addition, HTLV-1 expresses the *hbz* gene from the minus strand of the proviral genome that encodes HBZ, the helix basic zipper protein [14]. 

Tax has several activities that contribute to transformation [15]. Tax is a potent transcriptional activator of the viral promoter, and it is mediated through the cAMP response binding protein (CREB)/activated transcription factor (ATF) pathway [16]. This activity is the result of Tax binding to the CREB-binding protein (CBP) and p300, which are histone acetyl transferases [17]. Tax also activates canonical and non-canonical nuclear factor kappa B (NFκB) pathways [18]. The induction of canonical NFκB activity in HTLV-1-infected cells is the result of Tax interaction with the regulatory subunit of the kinase for inhibitor of NFκB (IκB kinase γ, IKKγ, or NFκB essential modulator (NEMO)) [19]. The induction of non-canonical NFκB signaling in HTLV-1 infected cells is the result of Tax interactions with NFκB-inducible kinase (NIK) and the NFκB2 precursor protein, p100 [20,21]. NFκB activation is required for the HTLV-1 immortalization of T-lymphocytes [22]. Other activities of Tax contributing to cell transformation have been described [23] 

However, Tax is also a key target of CD8+ CTL responses [24]. Tax is expressed in only 30–50% of uncultured ATLL samples due to CpG DNA methylation and inactivating histone methylation and acetylation marks of the 5′ half of the integrated proviral DNA [25,26]. In contrast, HBZ is universally expressed in ATLL samples, as well as in uncultured cells from asymptomatic and HAM subjects [27]. We and others hypothesized that Tax initiates tumor formation, whereas *hbz* RNA and HBZ protein are critical for proliferation and tumor cell maintenance [16,28].

## 4. HTLV-1 DNA Methylation

DNA methylation results from the covalent transfer of a methyl group from S-adenosyl-L-methionine to cytosine in CpG dinucleotides [29]. Promoter regions of genes have high GC contents, which are called CpG islands, and these may be methylated [30]. Methylation can change the activity of a DNA segment. When localized in a gene promoter, DNA methylation typically acts to repress gene transcription [31]. 

Most HTLV-1-infected cells in vivo harbor a provirus that is transcriptionally silent [32]. There are considerable data demonstrating that DNA methylation is at least partially responsible for transcriptional silencing, as described for other viruses. It is considered a host defense mechanism for inactivating retrovirus expression [33]. It is also recognized as a mechanism for virus-infected cells to escape host immune responses and to establish latency. The HTLV provirus is characterized by hypermethylation of the 5′ long terminal repeat sequence (LTR) and hypomethylation of the 3′ LTR in both latently infected cell lines and ATLL cells containing a complete provirus [25]. The differences in DNA methylation between the 5′ and 3′ LTRs suggest that this results in the repression of Tax with continued Hbz expression. The spreading of methylation from repetitive elements has been suggested to play a role in the epigenetic inactivation of some tumor suppressor genes [34]. In contrast, others have suggested that DNA methylation occurs first in the *gag*, *pol*, and *env* genes of the provirus, and then it extends in the 5′ and 3′ directions in vivo, and when the 5′ LTR becomes methylated, viral transcription is silenced [35]. Provirus gene expression is reactivated in latently infected cell lines by hypomethylating agents, such as 5-azacytidine [25]. Alternatively, transcription is rapidly recovered after in vitro culture of ATLL cells [36].

ATLL genomes are characterized by a prominent promoter-associated CpG island and DNA hypermethylation; the level of hypermethylation is correlated with poor prognosis [37]. Mutations in epigenetic regulators Tet methylcytosine dioxygenase 2 (TET2) and mixed-lineage leukemia protein 3 (MLL3) have been described in patients with ATLL [38]. However, ATLL prognosis is not correlated with mutations in the epigenetic regulators associated with hypermethylation in other cancers, which are commonly found in genes encoding TET2, isocitrate dehydrogenase 2 (IDH2), and DNA methyltransferase 3A (DNMT3A) [39]. CpG hypermethylation in ATLL cells is noteworthy in genes encoding zinc finger transcription factors and major histocompatibility class I proteins [37].

Watanabe et al. identified differentially methylated positions (DMPs) specific to HTLV-infected cells, and identified those that correlated with ATLL development and progression, including CpG islands in genes encoding thymocyte-expressed molecule (THEMIS), leukocyte-associated immunoglobulin-like receptor 1 (LAIR1), and ring-type E3 ubiquitin transferase 130 (RNF130), which negatively regulate T-cell receptor (TCR) signaling [40]. Other genes hypermethylated in ATLL included cyclin-dependent kinase inhibitor 2A (CDKN2A), Kruppel-like factor 4 (KLF4), and bone morphogenetic protein 6 (BMP6) [37,41,42]. They found that hypomethylating drugs had significant anti-ATLL activity in cell cultures and patient-derived xenograft mice. Other oncovirus-mediated cancers, including HPV-positive head and neck squamous cell carcinomas and EBV-positive gastric carcinomas, also exhibit aberrant DNA hypermethylation and susceptibility to DNA demethylating agents [43,44,45,46]. 

## 5. HTLV-1 Chromatin-Associated Histone Modifications

Post-translational modifications of histones play an important role in the epigenetic regulation of chromatin [47]. The most common modifications are the methylation of arginine or lysine residues, or the acetylation of lysine. Histone methylation involves the addition of one to three methyl groups. Two histone modifications are particularly associated with active transcription. Trimethylation of lysine 4 of histone 3 (H3K4me3) occurs at the promoter of active genes [48]. The trimethylation of lysine 36 of histone 3 (H3K36me3) occurs in the body of active genes. Three histone modifications are particularly associated with repressed genes. These include trimethylation of lysine 27 of histone H3 (H3K27me3), di- and tri-methylation of lysine 9 of histone 3 (H3K9me2/3), and trimethylation of lysine 20 of histone 4 (H4K20me3). These covalent modifications of histones act as a scaffold for the recruitment of specific regulatory proteins. Lysine acetylation eliminates a positive charge, thereby weakening the electrostatic interactions between histones and DNA, resulting in the partial unwinding of DNA and making it more accessible for gene regulation.

Transcriptional initiation, elongation, and mRNA processing are co-regulated through the posttranslational modification of the RNA Pol II (RNAPII) C-terminal domain (CTD). CTD is phosphorylated by transcription initiation factor IIH (TFIIH) and positive transcription elongation factor b (pTEFb), and bound by negative elongation factor A (NELF-A) and positive elongation factor Spt5 [49]. NELF-A colocalizes with RNAPII at chromatin insulators [50]. Moreover, polypyrimidine tract-binding protein (PTB) and histone modifications, such as H3K36me3, are also linked to CTD modification and RNAPII activity. 

Tax interacts with multiple histone-modifying enzymes. This includes histone acetylases (HATs) and deacetylases (HDACs). Tax binds histone acetylases, CREB-binding protein (CBP), and p300 in order to transactivate the viral promoter [51]. This was demonstrated in cultured cells, as well as within an in vitro reconstituted chromatin transcription system [52]. HDAC1 is recruited to the viral promoter by the cAMP response element-binding protein (CREB) [53]. Tax relieves transcriptional repression by promoting the phosphorylation of CREB, which releases HDAC1 from the viral promoter [54]. Tax interacts with histone deacetylase NAD-dependent deacetylase sirtuin-1 (SIRT1), which represses HTLV-1-associated viral gene expression, but this has no effect on NFκB-regulated genes [55]. This effect is a result of the SIRT1 inhibition of the Tax-mediated recruitment of CREB, CREB-regulated transcription coactivator 1 (CRTC1), and p300 to the HTLV-1 LTR. Resveratrol is a nutritional supplement that activates SIRT1, and it has inhibitory activity toward HTLV-1 gene expression in infected cells in culture. In contrast, the HDAC inhibitor, valproate, was shown to enhance HTLV-1 expression in patients [56]. Valproate-induced reactivation of the virus from latency resulted in a reduction in proviral loads several months after treatment, presumably through CTL clearance of infected cells [57].

Tax binds the histone methyltransferases SUV39H1 and SET and MYND domain-containing protein (SMYD3) [58]. Tax binding to SUV39H1 promotes its relocalization in the nucleus, resulting in a diffuse rather than a speckled distribution. Furthermore, SUV39H1 represses the transcriptional activity of Tax, suggesting a negative feedback loop [59]. SMYD3 promotes the di- and tri-methylation of H3K4. SMYD3 binding to Tax enhances its ability to activate NFκB. 

The suppression of incoming DNA by transcriptional silencing is an important mechanism, whereby invading viruses block the initiation of the cellular innate immune response. This has been shown for the murine leukemia virus [60] and many DNA viruses [61,62,63,64]. In all of these cases, the mechanism involves histone loading onto the viral DNA and their posttranslational modifications to silence gene expression. Unintegrated retroviral DNA acquires chromatin shortly after synthesis [65]. The chromatin is characterized by repressive epigenetic marks, such as H3K9me3, shortly after nuclear entry. Many viruses have evolved genes that function to inactivate the silencing mechanisms and to allow for virus expression. This includes Vpr of HIV-1 and Vpx of HIV-2 [66,67]. HTLV-1 Tax activation of NFκB results in a reduction of H3K9me3 marks on unintegrated HIV DNA and a significant increase in the activating H3Ac epigenetic mark [68]. The SMC5–SMC6 complex localization factor 2 (SLF2) is recruited to HIV unintegrated DNA to compact viral chromatin marks [69]. The mechanisms underlying these observations remain to be clarified.

H3K27me3 is a mark of chromatin condensation and gene silencing, and it is regulated by enhancer of zeste homolog 1 (EZH1) and EZH2, which are core components of polycomb repressive complex PRC2 with embryonic ectoderm development protein (EED) and suppressor of zeste 12 homolog (SUZ12). Several drugs have been identified as EZH1 or EZH2 inhibitors, including the FDA-approved EZH2 inhibitor tazemetostat [70]. H3K27me3-mediated gene repression correlates with poor prognosis in ATLL [71,72]. The long noncoding RNA, ANRIL (antisense noncoding RNA in the INK4 locus), promotes EZH2 and NFκB activation in ATLL cells [73]. The combined inhibition of EZH1/2 resulted in the lethality of ATLL and other lymphoma cell types in culture [74]. A phase 1 clinical trial with this drug is underway [75].

H2AK119ub1, a signature of PRC1 resistance, is enriched on the latent HTLV-1 provirus [76]. It is induced by p38-MAPK signaling. Deubiquitylation inhibits plus-strand transcription, the modification of histones at genes for p300, SET domain-containing 2 (SET2), lysine demethylase 6A (KDM6A), and switch/sucrose non-fermentable complex (SWI/SNF)-mediated chromatin remodeling, which are known to be frequently altered in human cancers, including other T-cell neoplasms [76].

In summary, the inhibition of histone-modifying enzymes is an active area of investigation in ATLL and other T-cell lymphomas.

## 6. Nucleosome Positioning along the HTLV-1 Provirus

Nucleosomes are composed of an octomer of histone proteins, with two copies each of the histones H2A, H2B, H3, and H4 [77]. Linker histones, such as H1, sit at the base of the nucleosome near the DNA entry and exit binding sites [78]. DNA is compacted into nucleosomes, and then it is folded into more complex structures to form chromosomes. Histone modifications may “loosen” nucleosome core–DNA associations. Nucleosome-remodeling enzymes slide nucleosomes along the DNA and disrupt histone–DNA contacts. 

Tax induces the clearance of nucleosomes from the HTLV-1 promoter [79]. Nucleosome depletion depends on CBP/p300 HAT activity. The histone chaperone, nucleosome assembly protein 1 (Nap1), contributes to this activity, which is thought to be involved in nucleosome disassembly [80]. Tax also displaces the SWI/SNF chromatin-remodeling complex [79]. Both Tax and HBZ bind to the SWI/SNF-related matrix-associated actin-dependent regulator of the chromatin subfamily A member 4 (SMARCA4) component of the SWI/SNF remodeling complex [80,81,82]. Although some studies suggested an important role of SMARCA4 in Tax-mediated transcriptional transactivation [81,82], another found it to be dispensable for Tax-mediated transcriptional transactivation [83]. FACT (facilitates chromatin transcription) proteins (SUPT16H and SSRP1) also play a role in nucleosome assembly and disassembly to facilitate the sliding of RNA polymerases [84]. They play an undefined role in the inhibition of HTLV-1 transcription. MicroRNAs (miRNAs) may also modulate chromatin function, and the levels of several miRNAs are altered in HTLV-1-infected cells [85]. However, their role in chromatin remodeling and HTLV-1 biology remains unclear. 

## 7. Topology-Associated Domains in HTLV-1-Infected Cells

Studies of ATLL DNA and chromatin modifications demonstrated a sharp boundary in the provirus near nucleotide 6718 (NC_001436.1), with the 5′ portion, but not the 3′ portion, of the genome being silenced in most patient samples (Figure 1). This suggests that a chromatin insulator may account for this transition from tightly packed heterochromatin, to lightly packed euchromatin [86]. The histones in the heterochromatic domains are hypoacetylated and lack methylation of H3K4, but they are enriched in methylation of H3K9 and H3K36 and are associated with specific chromodomain-containing proteins, leading to the condensation of chromatin and the inhibition of transcription initiation and elongation [86]. 

An insulator is a genetic boundary element that blocks interactions between enhancers and promoters [87]. The best characterized mammalian insulator element is bound by a transcriptional repressor, CTCF (i.e., 11-zinc finger CCCTC-binding factor). CTCF binds together strands of DNA, forming chromatin loops, and it anchors DNA to cellular structures like the nuclear lamina [88]. CTCF binding can promote or repress gene expression, but whether this is facilitated through its looping activity is unknown.

CTCF binds 55,000 DNA sites in 19 diverse cell types [89] to multiple sequences using various combinations of its zinc fingers as a multivalent protein. CTCF binding sites (CTCF-BS) act as nucleosome positioning anchors to align various genomic signals. CTCF forms homodimers for looping activity. CTCF-BS borders 80% of chromatin loops. Other chromatin-associated proteins at loop boundaries include PRC1, adapter protein-1 (AP-1), and Ying Yang 1 (YY1) [90,91,92]. CTCF also interacts with YY1 and cohesin to stabilize repressive loops. The resulting chromatin loops of 100 kb–2 Mb control gene expression by regulating contacts between enhancers and promoters [93]. Cohesins include components such as Structural Maintenance of Chromosomes 1 (SMC1) and 3 and SCC1 (double-strand-break repair protein Rad21), and chromatin loading is regulated by SCC2, establishment of sister chromatin cohesion N-acetyltransferase 1 (ESCO1), and cohensin subunit-2 (SA-2). Cohesin keeps sister chromatids connected during metaphase so that they segregate to opposite poles, and it facilitates spindle attachment and DNA repair by recombination. Cohesin is also responsible for transcriptional regulation, DNA-break repair, chromosome condensation, non-homologous centromere coupling, chromosome architecture and rearrangement, DNA replication, and activities during meiosis. The disruption of chromatin loops deregulate gene expression and cause developmental abnormalities or cancer [94,95]. CTCF also interacts with chromatin remodelers, such as chromodomain helicase DNA binding protein 4 (Chd4) and (SMARCA5) [96].

CTCF regulates the expression of many DNA viruses. EBV has 19 CTCF-BS, with most of them at key regulatory regions (Cp, Qp, and LMP1/2 promoters) [97]. With the removal of CTCF binding from the Qp region, repressive chromatin marks spread into the Qp start site and silence transcription. CTCF binding also negatively regulates Cp activity in type 1 latency infected cells. Deletion of the CTCF-BS from the latency-associated nuclear antigen (LANA) promoter disrupts cohesin binding and DNA loop formation between genes for latency membrane protein 1 (LMP1)/LMP2A and origin of replication OriP, and loss of episome stability [98]. CTCF was detected at 17 sites in the KSHV genome, including three sites in the latency control region. Disruption of these sites led to a loss of episome stability and deregulation of viral gene expression. A CTCF-BS in KSHV regulates mRNA production, RNAPII programming, and nucleosome organization of the latency transcript control region. Binding of cohesin, but not CTCF, is required for the repression of KSHV immediate early transcription [99,100,101]. These findings suggest that cohesin functions coordinately with CTCF to regulate the switch between latent and lytic gene expression. Glycyrrhizic acid (GA), a product from licorice root, inhibits KSHV-induced proliferation of cells [50]. GA binds to cohesin components, pauses RNAPII at CTCF–cohesin–BS, inhibits transcription, and derails sister chromatid cohesion. 

Herpes simplex virus type 1 has seven CTCF-BS, including three sites within control locus 2 (CTRL2) downstream of the latency-associated transcript (LAT) enhancer [102]. Mutations within these CTCF-BS disrupt the epigenetic regulation of HSV-1, which is critical for reactivation of the virus from latency. In HCMV-infected cells, CTCF negatively regulates major immediate early gene expression during primary infection through a binding site in intron A [102]. In contrast, in adenovirus infection, knockdown of CTCF suppresses viral DNA replication as well as late, but not early, gene expression [103].

The boundary between endogenous retroviruses and adjacent genes is enriched in CTCF-BS [104]. CTCF silencing of retroviral vectors has also been studied in order to prolong the duration of gene expression. In fact, DNA methylation was proposed to have evolved as a cellular mechanism to silence retroviral elements, preventing the spread of transposable elements through the genome.

A single conserved CTCF binding site (vCTCF-BS) has been identified in the integrated HTLV-1 provirus, flanking methylated proviral DNA and closed chromatin in the 5′ provirus and unmethylated DNA and open chromatin in the 3′ provirus [105]. Miura et al. reported that loss of CTCF binding to this site did not affect epigenetic modifications of the HTLV-1 provirus [106]. However, Cheng et al. found that mutation in the vCTCF-BS disrupted epigenetic barrier function, resulting in enhanced DNA CpG methylation downstream of this site on both strands of the integrated provirus and H3K4me3, H3K36me3, and H3K27me3 chromatin modifications up- and down-stream of the site [107]. A majority of clonal cell lines infected with wild-type HTLV-1 exhibited increased plus-strand gene expression with CTCF knockdown, while expression in mutant HTLV-1 clonal lines was unaffected (Figure 2). Melamed et al. found that DNA loops were formed between the vCTCF-BS and binding sites in the immediately flanking host genome and clone-specific transcription in *cis* at non-contiguous loci up to >300 kb from the integration site [108]. These findings suggest that CTCF binding to the HTLV-1 genome regulates virus and host cell gene expression.

## 8. Future Queries

The study of the epigenetic regulation of HTLV-1 is a burgeoning field of investigation. Many questions remain to be answered. What are the dynamics of the epigenetic modulation of the HTLV-1 genome before and after integration? Are there differences in epigenetic alterations in malignant compared to non-malignant infected cells? What is the importance of epigenetic alterations reported in cell culture studies to those present in vivo in animal model studies and patients? Does epigenetic regulation differ in asymptomatic infected individuals compared to the patients with HAM, ATLL, or other inflammatory diseases? How does epigenetic regulation of the provirus and the cellular genome differ in ATLL cases wherein the Tax gene is intermittently expressed and those cases in which it is deleted or permanently silenced? Are there key cellular genes that determine pathogenesis that are regulated by epigenetic alterations in HTLV-associated disease? Are there specific therapeutic approaches that disrupt epigenetic regulation in HTLV-1-infected cells? Can such maneuvers have a role in the treatment or prevention of diseases? Can one reduce the proviral load in infected individuals? Do approaches for disrupting or reinforcing latency that are being tested in HIV-1-infected individuals have applications in HTLV-1 disease? How does CTCF regulate the integrated provirus and the cellular genome in cultured cells and in vivo? Does it work primarily as a transcriptional repressor, or do its activities in promoting chromatin loops or interactions with the nuclear laminin contribute to its activity? Do interactions with cohesin components contribute to the role of CTCF? Are there chromatin insulators, other than CTCF, contributing to the regulation of HTLV-infected cells? Answers to these and other questions should provide novel insights into our understanding of this complex retrovirus.

## 9. Summary

HTLV-1 is a complex retrovirus associated with ATLL through the actions of Tax and HBZ oncoproteins as well as through the integration of site-specific effects on the host cell. Alterations in DNA methylation, histone modifications, and DNA looping play key roles in modifying viral and host cell gene expression. Future studies will uncover the molecular underpinnings of these observations and their pathogenic effects.

## Figures and Tables

**Figure 1 microorganisms-10-00084-f001:**
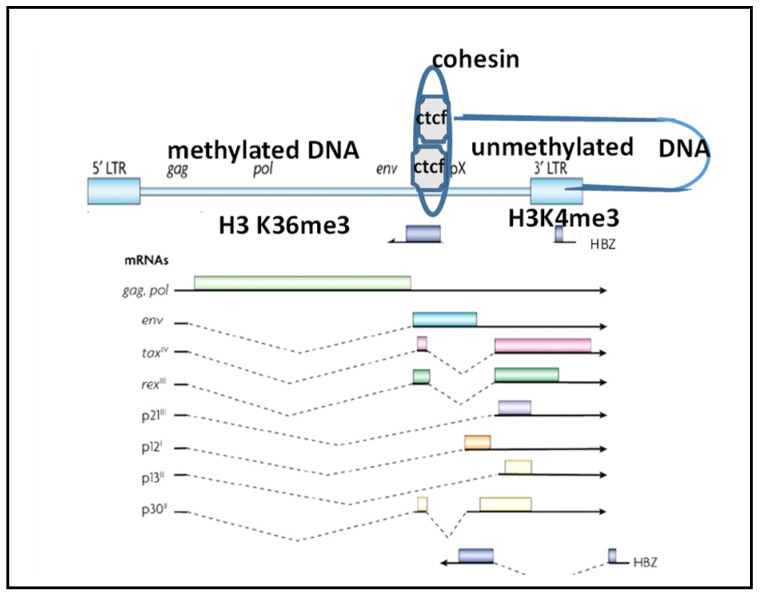
Schematic diagram of the HTLV-1 provirus, CTCF-BS, epigenetic modifications, and transcripts. The top figure shows potential looping of the HTLV-1 CTCF-BS with a cellular CTCF-BS, with bound cohesin. The epigenetic barrier created by CTCF results in 5′ provirus methylated DNA and 3′ provirus unmethylated DNA as well histone marks of inactive and active chromatin, respectively.

**Figure 2 microorganisms-10-00084-f002:**
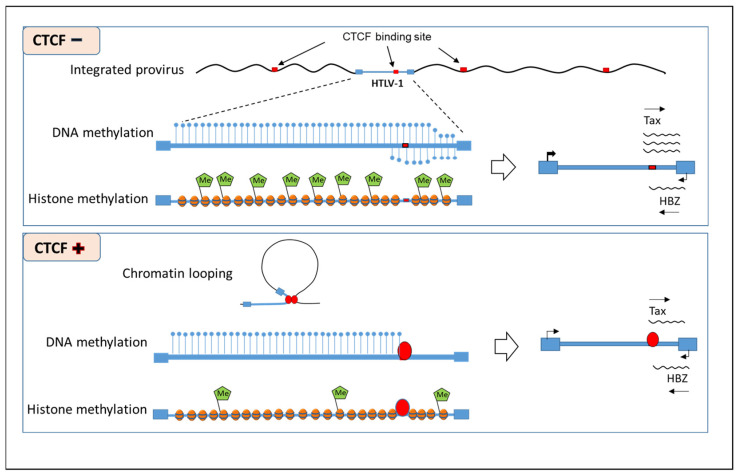
CTCF is a critical boundary element that regulates epigenetic modifications of HTLV-1. CTCF prevents the extension of DNA methylation in the pX and 3′ LTR regions on both the sense and antisense strands of the HTLV-1 provirus. The interaction of CTCF and proviral DNA also leads to a lower level of methylation for histone 3, both up- and downstream of the CTCF binding site of the provirus. As a result of these epigenetic modifications, less Tax is expressed.

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
