# Peer review of "Epigenetic Regulation of Human T-Cell Leukemia Virus Gene Expression"

_microorganisms, 2021, doi:10.3390/microorganisms10010084_

Round 1
Reviewer 1 Report
The manuscript from Lee Ratner is a comprehensive review of the different epigenetic mechanisms that regulate the expression of HTLV-1 in human host cells. The manuscript is really well written. Although the complexity of the subject, the reading is fluid. The different associated diseases are mentioned, the epigenetic mechanisms are extensively described in paragraphs and the references are updated. I didn't find I have not found major concerns. The only aspect that can be implemented with more statements and references is on the control of treatments to potentially inhibit NF-kB pathways in ATL or in vitro infection models.
Author Response
No changes requested
Reviewer 2 Report
This manuscript by Ratner, is a review of epigenetic mechanisms regulating the expression of the human T-cell leukemia virus (HTLV). The integral role of the integrated HTLV-1 provirus in the multifold pathogenic outcomes of HTLV-1 infection highlights the critical but still relatively poorly understood role for epigenetic modulation of HTLV-1 gene expression in the critical distinctions of whether or not infected individuals develop any disease phenotype, and if so, which one. Thus, there is value and significance for careful reviews of this area. This review is also timely in view of some recent new observations by Ratner and others related to epigenetic and chromatin regulation of HTLV, as well as more general work integrating evolving principles of epigenetics into the biological contexts of gene expression. This review does a particularly nice job integrating a flurry of older papers (early 2000’s) on DNA and histone modifications and nucleosome organization in the HTLV LTR, with more recent work on the CTCF site and more recent clinical work on inhibitors of DNA/histone modifying enzymes. While relatively succinct, the review is comprehensive of both earlier and more recent work.
The distinction between a good review and one that helps drive a field forward often revolves around the clear articulation of critical unanswered questions and future directions for the field. Some important remaining questions are in fact scattered through the text. However, while this is a reasonable review in its current format, the addition of a single, thoughtful paragraph at the end outlining a handful of these key questions and potential future directions (such as the relationship of epigenetic regulation of the sense versus anti-sense promoters, implications for pathogenesis of HAM/TSP as opposed to ATLL and nature of cells in asymptomatic infection, more direct considerations of the effects of epigenetic changes on clearance of HTLV-1 infections cells (i.e recent discussions of applicability of “kick and kill” to HTLV-1, alluded to in the Discussion of histone methyltransferase inhibitors in HAM/TSP, to name only a few of perhaps a half-a-dozen really interesting and provocative questions), would markedly add to the interest and long-term significance of this review.
Minor Points:
- Lines 74, 75 describes “We hypothesize that Tax initiates tumor formation whereas hbz RNA and HBZ protein are critical for proliferation and tumor cell maintenance” is a reasonable statement, however it is worth noting (i.e. worth referencing) that this hypothesis is by no means unique to this review and is stated more or less explicitly by several earlier reviews and research reports several authors including Giam, Matsuoka and others as well as by the Ratner group and collaborators.
- In the methylation section (lines 93-95) it would be helpful to make the implied connection of 5’LTR methylation and immune escape mechanisms for Tax expression but not HBZ more explicit, as this is likely to be a key role/selective pressure for silencing.
- Lines 169-171 discussion of Tax effects are ambiguous, just referring to LTR. In fact, the paper in question is referring to effects on the HIV LTR not the HTLV-1 LTR. As written, these sentences could generate confusion as to which unintegrated provirus was shown to be regulated in this way. The next lines,172-173 and reference 76 are also about HIV DNA. It does raise the interesting question of whether anything is known about nucleosome/histone status of unintegrated HTLV-1 DNA?
- Lines 228-234 seem a bit out of place in the flow of the review. There is no apparent relationship drawn between the information in this paragraph about basic mechanisms of transcription and HTLV regulation and the preceding and following paragraphs on CTCF nor is any relationship between the material in this paragraph and HTLV-1 regulation presented.
Author Response
see attached word file
